# Systemic Immune-Inflammation Index Predicted Short-Term Outcomes in Patients Undergoing Isolated Tricuspid Valve Surgery

**DOI:** 10.3390/jcm10184147

**Published:** 2021-09-14

**Authors:** Jungpil Yoon, Jaewan Jung, Youngick Ahn, Jimi Oh

**Affiliations:** 1Department of Anesthesiology and Pain Medicine, Asan Medical Center, University of Ulsan College of Medicine, Seoul 138736, Korea; wizdumb@naver.com; 2Department of Internal Medicine, Wonkwang University Hospital, Iksan 54538, Korea; alexanderx@naver.com; 3Department of Anesthesiology and Pain Medicine, Ulsan University Hospital, University of Ulsan College of Medicine, Ulsan 44033, Korea; 0735495@uuh.ulsan.kr

**Keywords:** systemic immune-inflammation index, tricuspid valve, postoperative complications, prognosis

## Abstract

Systemic immune-inflammation index (SII, platelet × neutrophil/lymphocyte ratio) has recently been identified as an inflammatory marker. We aimed to evaluate the prognostic implications of preoperative SII in patients undergoing isolated tricuspid valve (TV) surgery. In total, 213 patients who underwent isolated TV surgery between January 2000 and December 2018 were enrolled. They were divided into two groups, as follows: low SII (<455.6 × 10^9^/L), and high SII (≥455.6 × 10^9^/L). The correlation between SII and clinical outcomes was analyzed via the Cox regression and the Kaplan–Meier analyses. The primary outcomes considered were all-cause mortality and major postoperative complications within a 30-day period after isolated TV surgery, including major adverse cardiovascular or cerebrovascular events, pulmonary and renal complications, stroke, sepsis, multi-organ failure, wound, and gastrointestinal complications. In total, 82 (38.5%) patients experienced postoperative complications. Multivariable analyses revealed that high preoperative SII values were independently associated with the major 30-day postoperative complications (hazard ratio 3.58, 95% confidence interval 1.62–7.95, *p* = 0.001). Additionally, Kaplan–Meier analysis revealed that the probability of undergoing major 30-day postoperative complications was significantly elevated in patients with high versus low SII values (*p* < 0.001). These results indicate that SII, a readily available parameter, is significantly associated with poor outcomes in patients undergoing isolated TV surgery.

## 1. Introduction

Systemic inflammation plays a central role in cardiovascular disease. Circulating neutrophils and platelets with roles in chronic inflammatory processes and lymphocytes that function throughout subsequent immune responses are important contributors to atherosclerosis [1,2]. Inflammation and immune-based prognostic scores, such as neutrophil/lymphocyte ratio (NLR), platelet lymphocyte ratio (PLR), as well as established pro-inflammatory biomarkers (leukocytosis and C-reactive protein levels) have been shown to predict mortality and morbidity after cardiac surgery [3,4]. Recently, systemic immune-inflammation index (SII), which is a combined indicator of neutrophil, platelet, and lymphocyte levels, has emerged as a prognostic factor for cardiovascular disease. High SII scores have previously been shown to indicate poor prognoses in patients with coronary artery disease, myocardial infarction, and chronic heart disease [5,6,7,8]. However, the association between preoperative systemic inflammation and prognosis of isolated tricuspid valve (TV) surgery has not been reported. Indeed, patients undergoing isolated TV surgery are frequently exposed to chronic inflammation because symptom-based surgical delays, advanced heart failure, and comorbidities, including diabetes, hypertension, hepatic dysfunction, and prior left-side valve surgery, contribute to chronic inflammation state [9]. In addition, isolated TV surgery has been identified as a high-risk surgery with a 10% in-hospital mortality rate and poor long-term outcomes regardless of the improvement of surgical techniques or postoperative management [10,11]. Therefore, efforts need to be made to provide prognostic factors to predict postoperative clinical outcomes, thereby offering appropriate patient selection for surgery.

We hypothesized that systemic inflammation, evaluated by SII, may be an important prognostic factor of postoperative outcomes in patients undergoing isolated TV surgery. Therefore, this study aimed to assess the prognostic value of preoperative SII for predicting postoperative outcomes of isolated TV surgery.

## 2. Materials and Methods

### 2.1. Study Design and Participants

This single-institutional retrospective cohort study was approved by the ethics committee of our institution (AMC IRB 2020-1580). Due to the observational nature of the study, the need for written informed consent was waived. The following patients were included in the study: (I) adults aged >18 years (II) who underwent isolated TV surgery between January 2000 and December 2018 at our institute. Exclusion criteria were as follows: (I) clinical evidence of acute or chronic infection, (II) other surgery performed concurrently, and (III) incomplete medical records. Clinical data, including demographics, comorbidities, preoperative laboratory findings, medication use, postoperative complications, and mortality, were collected via a retrospective review of the computerized patient record system (Asan Medical Center Information System Electronic Medical Record) and the Asan Medical Center Cardiovascular Surgery and Anesthesia Database [12]. This study was completed according to the Strengthening the Reporting of Observational Studies in Epidemiology statement [13].

### 2.2. Markers of Systemic Inflammation

In all patients, complete blood count (CBC) was routinely measured with an auto-analyzer within the 1-week period that preceded isolated TV surgery. Systemic inflammation was assessed based on NLR and PLR levels obtained via routine CBC analysis. NLR was calculated by dividing the neutrophil count by the lymphocyte count. Similarly, the PLR was calculated by dividing the platelet count by the lymphocyte count. SII was calculated using the following formula: SII = platelet count × neutrophil count/lymphocyte count [14].

### 2.3. Outcomes and Definitions

The primary outcome assessed was a composite of major complications and all-cause mortality throughout the 30-day period that followed isolated TV surgery. Major postoperative complications were designated based on European Perioperative Clinical Outcome definitions [15], as follows: (a) major adverse cardiovascular events (e.g., myocardial infarction, malignant ventricular arrhythmia, hospitalization because of heart failure, and application of a mechanical assist device); (b) pulmonary complications (acute lung injury or pneumonia for any reason and acute respiratory distress syndrome, requiring prolonged mechanical ventilation or tracheostomy); (c) renal complications; (d) stroke; (e) sepsis; (f) multi-organ failure; (g) wound complications that required reoperation; or (h) gastrointestinal complications. Secondary outcomes included duration of mechanical ventilation or inotropic support following surgery, length of intensive care unit and hospital stay, and readmission for any reason within 30 days post surgery.

### 2.4. Statistical Analyses

Descriptive analyses are presented as number (proportion), median (inter-quartile range), or mean ± standard deviation, as appropriate. Participant characteristics and postoperative clinical outcomes were compared using chi-square or Fisher’s exact tests for categorical variables and the Student *t*-test or Wilcoxon rank-sum test for continuous variables. The optimal SII cut-off point was determined by analyzing a composite of major 30-day postoperative complications. The SII cut-off value (455.6 × 109/L) was defined by analyzing receiver operating characteristic (ROC) curves of patients using the Youden index method with and without an endpoint of major 30-day postoperative complications. The method used to determine this cut-off was based on previous reports [16,17].

Patients were divided into two groups according to SII, as follows: low SII (<455.6 × 109/L) and high SII (≥455.6 × 109/L). Univariate and multivariable analyses were performed using the Cox proportional hazards regression model. Factors related to major 30-day post-operative complications were identified via univariate analysis. Possible prognostic factors, including variables with *p*-values < 0.05 via univariate analyses, were incorporated into the stepwise multivariable Cox regression model [18]. Prognoses and major complication-free survival rates of patients of each SII group were compared via Kaplan–Meier analysis, and significance was assessed using the log-rank test. To test the robustness of our results, sensitivity analysis was performed using different cut-off values mentioned in previous studies and the median value in our study population [5,6,7,8,16]. We used the Cox proportional hazards models and the Kaplan–Meier curves to identify relationships between prognostic factors and major 30-day postoperative complications. All statistical analyses were performed with the Statistical Package for the Social Sciences version 21.0 (IBM Corporation, Armonk, NY, USA). Two-sided *p*-values < 0.05 were considered statistically significant.

## 3. Results

### 3.1. Baseline Characteristics According to SII

A total of 250 patients undergoing isolated TV surgery were enrolled in this study. After excluding patients with a history of acute or chronic infection (e.g., infective endocarditis) (*n* = 5), with incomplete data within one month before surgery (*n* = 5), or who had undergone concomitant surgery at this time (*n* = 27), the final study cohort was 213 patients (Figure 1). Of the 213 patients included in the study, 77 were men (36.2%), and 136 were women (63.8%). The average age of study participants was 57.2 ± 12.8 years.

The cut-off value of SII (455.6 × 10^9^/L) was calculated by analyzing the ROC curve between patients with and without major postoperative complications (Figure 2). We divided patients into low SII (<455.6 × 10^9^/L) and high SII (≥455.6 × 10^9^/L) groups based on whether their SII scores were greater or less than the cut-off value determined. Baseline and clinical characteristics of patients based on SII scores are summarized in Table 1. Patients with higher SII scores were more likely to have lower preoperative hematocrit and albumin levels than those with lower SII scores. The two groups of patients did not differ when other baseline and operative characteristics were compared (Table 1).

### 3.2. Clinical Outcomes

Postoperative early clinical outcomes are summarized in Table 2. Composite 30-day major complications occurred in 82 patients (38.5%), and rates of complications were higher in the high than in the low SII group (83.3% (50/60) versus 20.9% (32/153), *p* = 0.001). Moreover, 19 (31.7%) of 153 patients with high baseline SII scores had major adverse cardiovascular events (MACE), and 14 (23.3%) experienced pulmonary complications. Patients with low baseline SII scores had a significantly lower incidence of MACE (9.8%) and pulmonary complications (6.5%) than those with high baseline SII scores during the 30-day follow-up period. Regarding secondary outcomes, the duration of hospital stay (28.2 ± 27.6 days versus 13.3 ± 18.2 days, *p* = 0.001) was significantly longer for patients with high versus low SII scores. However, when 30-day readmission rates (6.7% (4/60) versus 6.5% (10/153), *p* = 0.717) and durations of intensive care unit stays (146.9 ± 257.6 days versus 101.1 ± 332.5 days, *p* = 0.286) were compared, no between-group differences were observed.

### 3.3. Prognostic Impact of SII

Results of the univariate analysis indicated that age, EuroSCORE, preoperative hematocrit, serum albumin, C-reactive protein, alanine aminotransferase, PLR, NLR, SII, and calcium channel blocker were prognostic factors for major 30-day postoperative complications (Table 3). After a multivariable analysis of the variables was performed, SII (hazard ratio (HR) 3.58, 95% confidence interval (CI) 1.62–7.95, *p* = 0.001) remained independent prognostic indicators of major 30-day postoperative complications.

The Kaplan–Meier analysis revealed that the probability of undergoing major 30-day postoperative complications was significantly elevated in patients with higher SII versus lower SII scores (*p* < 0.001, Figure 3).

### 3.4. Sensitivity Analyses of SII Cut-Off

At each SII cut-off value selected for sensitivity analyses, we observed a significant association between high preoperative SII scores and major 30-day postoperative complications (SII 1423.12 × 10^9^/L as the cut-off value, HR 2.63, 95% CI 1.18–5.84, *p* = 0.018, Appendix A) (SII 878.06 × 10^9^/L as the cut-off value, HR 2.19, 95% CI 1.05–4.54, *p* = 0.036, Appendix A) (SII 750.00 × 10^9^/L as the cut-off value, HR 2.81, 95% CI 1.37–5.76, *p* = 0.005, Appendix A) (SII 694.03 × 10^9^/L as the cut-off value, HR 3.97, 95% CI 1.72–9.19, *p* = 0.001, Appendix A) (SII 551.00 × 10^9^/L as the cut-off value, HR 2.12, 95% CI 1.11–4.08, *p* = 0.024, Appendix A) (SII 326.26 × 10^9^/L as the median value in our study population, HR 2.13, 95% CI 1.17–3.89, *p* = 0.014, Appendix A). In addition, on Kaplan–Meier analysis, the probability of undergoing composite major postoperative complications within 30 days was significantly higher in patients with high preoperative SII scores (log-rank test, *p* = 0.001, Appendix A).

## 4. Discussion

In this study, we assessed the prognostic value of SII for predicting postoperative short-term morbidity and mortality in 213 patients who underwent TV surgery. The main finding was that preoperative systemic inflammation evaluated by SII was significantly associated with an increased risk of postoperative composite complications and all-cause mortality within a one month after TV surgery.

Isolated TV surgery has a high rate of early postoperative mortality due to delayed surgical intervention, advanced heart failure symptoms, and right ventricle (RV) dysfunction [19]. Previous studies have pointed out that the poor prognosis of TV surgery was due to the patient’s aggravated conditions rather than the type of surgery or etiology of TR [10,11,20,21,22]. However, current guidelines for TV surgery recommend symptom-guided surgical intervention and do not suggest the risk stratification for patients undergoing isolated TV surgery [23]. In addition, EuroSCORE and STS score have not been validated in the field of isolated TV surgery because factors associated with increased mortality of tricuspid operation are not reflected in conventional scoring systems. In this regard, several studies have reported that preoperative anemia, renal/hepatic dysfunction, RV dilation, and significant postoperative TR are important when predicting long-term outcomes following isolated TV surgery [20,21,24,25]. In addition to these prognostic factors, we found that a high preoperative SII was significantly associated with postoperative complications in TV surgery.

Systemic inflammation and immune activation are previously identified independent prognostic factors for poor outcomes of chronic heart failure [26]. Moreover, Seo et al. reported that high SII predicted worse outcomes in chronic heart failure [8]. Recently, several studies have reported the prognostic value of SII in patients with coronary artery disease and infective endocarditis [6,16,27]. A retrospective study of patients undergoing off-pump coronary artery bypass grafting demonstrated that high preoperative SII scores are the only hematologic biomarker of postoperative complications [7]. Our results are consistent with those of previous studies and further expand our understanding of patients undergoing isolated TV surgery. Our study shows that SII is an independent biomarker of poor postoperative outcomes in patients undergoing isolated TV surgery. These results indicate that high SII (≥455.6 × 10^9^/L) is associated with an increased acute postoperative complication incidence.

Patients with high preoperative SII scores typically display thrombocytosis, neutrophilia, and lymphopenia, suggesting that there is a nonspecific combination between inflammation and impairment associated with the adaptive immune response [28]. Previous studies have demonstrated that chronic inflammation leads to protein metabolism changes or the decreased capacity to cope with acute surgical stress in patients undergoing major non-cardiac surgery, which results in a decreased ability to recover post surgery when compared with healthy patients [29].

Generally, neutrophils regulate inflammatory responses and secrete inflammatory mediators. Hyperactivation of the inflammatory response can potentially detrimentally affect myocardial function due to maladaptation. This can lead to cardiac metabolic disorders, negative inotropic effects, ventricular dysfunction, and myocardial remodeling. Thus, neutrophils are positively associated with inflammation and have been used to evaluate the diagnosis and prognosis of diseases [30,31,32]. In contrast, lymphocytes exhibit a regulatory response of the immune system. Interestingly, chronic inflammation, oxidative stress, and neurohormonal activation in patients with chronic heart failure increase the plasma cortisol levels and the catecholamine release, leading to the downregulation of the lymphocyte differentiation and proliferation, which further promotes lymphocyte apoptosis [33]. Previous studies have shown that low lymphocyte count is an early marker of physiologic stress [34,35]. Regarding platelets, in patients with systemic inflammation, disease may be exacerbated via a cascade of events, including platelet aggregation, adhesion, recruitment, and release [36]. Increased platelets have also been reported to be associated with the poor outcome in cardiovascular disease [37]. In this regard, our study suggests that high preoperative SII scores in patients undergoing isolated TV may lead to poor clinical outcomes after surgery and diminish a patient’s capacity to tolerate postoperative complications. However, it did not identify the precise mechanism by which high preoperative SII and poor clinical outcomes were associated. Therefore, further research to elucidate this mechanism is needed.

This is the first study to assess use of SII for predicting poor postoperative outcomes in patients undergoing isolated TV surgery. However, it has some limitations. First, optimal cut-off values for defining high SII scores in isolated TV surgery have not yet been established. In our study, the cut-off value for the high SII was arbitrarily determined using ROC, in accordance with previous studies [16,17]. Therefore, it is possible that other SII cut-off values may be more suitable for defining high SII scores. The heterogeneity of SII cut-off values described in the literature may impede use of SII in clinical settings. Thus, further studies are required to evaluate the optimal threshold of SII value in patients undergoing isolated TV surgery. Second, this study evaluated SII only once before isolated TV surgery and did not dynamically assess SII throughout the hospital stay. Thus, it is unclear whether alterations in SII scores may inform the care of patients undergoing isolated TV surgery. Third, although we adjusted for various factors known to influence major 30-day postoperative complication rates after isolated TV surgery, the retrospective nature of this study may have masked unknown confounders or prognostic factors. In particular, inotropic agents, ventilator parameters, fluid administration, and other intraoperative events that may significantly influence major 30-day postoperative complication occurrence were not included in our analysis. Therefore, our results should be interpreted with caution. Finally, this was an observational, single-institution study. Further studies of a large cohort are warranted to validate our results.

Despite its limitations, we believe our results have significant clinical implications. This is the first study to describe the utility of SII as a preoperative risk-assessment tool in patients undergoing isolated TV surgery. Regardless of the presence of RV dysfunction or baseline comorbidities, high SII scores were associated with increased rates of major acute postoperative complications. Therefore, SII scores can be used to perform risk stratification in patients undergoing isolated TV surgery. Finally, systemic inflammation is an adjustable risk factor, and high SII scores can be used to facilitate the determination of appropriate anti-inflammatory treatments, which may improve surgical outcomes of TV surgery in the future.

## 5. Conclusions

High preoperative SII scores, which may reflect the balance between immune and inflammation pathways, were associated with increased major 30-day postoperative complication risk after isolated TV surgery. SII can be easily calculated after routine analysis of differential white blood cell and CBC counts, suggesting that it is a simple and inexpensive biomarker of short-term outcomes of TV surgery. In addition to traditional risk scoring systems, SII may provide additional information about risk stratification and patient selection for TV surgery.

## Figures and Tables

**Figure 1 jcm-10-04147-f001:**
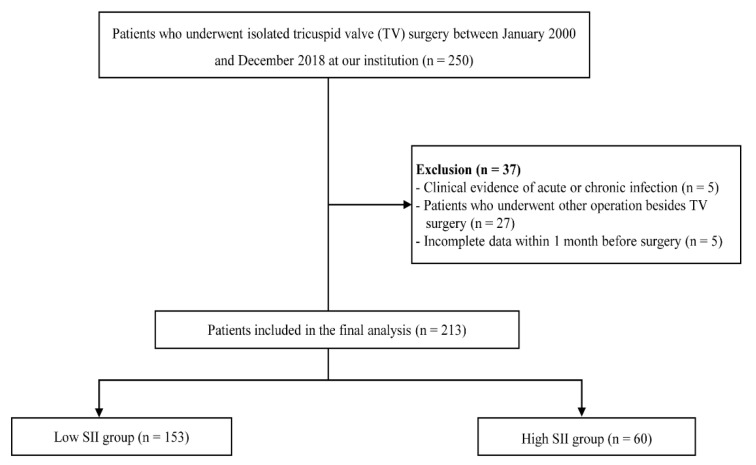
Flowchart of the patient selection and classification. SII, systemic immune-inflammation index.

**Figure 2 jcm-10-04147-f002:**
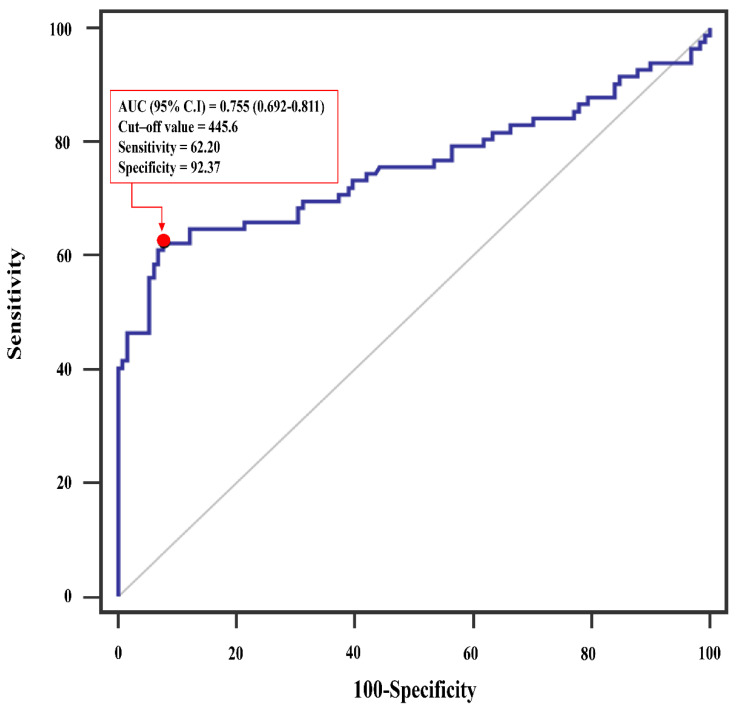
Receiver operating characteristics (ROC) analysis showing the prediction of major postoperative complication via systemic immune-inflammation index after isolated tricuspid valve surgery. AUC, area under the curve; CI, confidence interval.

**Figure 3 jcm-10-04147-f003:**
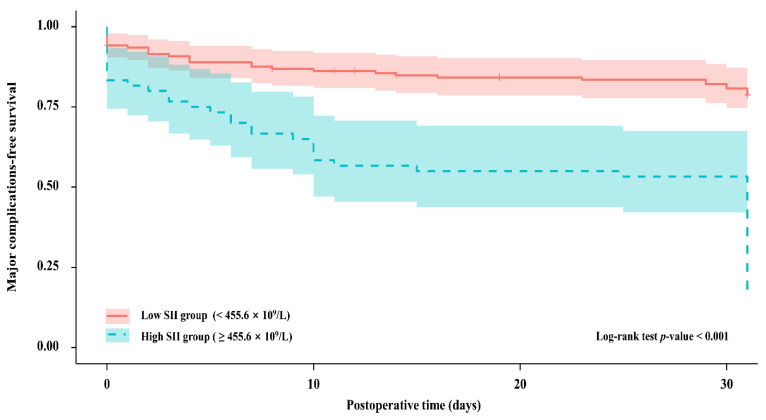
Kaplan–Meier curves for the probability of major postoperative complications between patients with different SII groups; SII, systemic immune-inflammation index; Low SII group, red; High SII group, blue.

**Table 1 jcm-10-04147-t001:** Baseline and operative characteristics of study participants.

Variables	Total	SII <455.6 × 10^9^/L	SII ≥455.6 × 10^9^/L	*p*-Value
N	213	153	60	
Demographic variables				
Age, years	57.2 ± 12.8	57.1 ± 12.9	57.5 ± 12.7	0.857
Female sex, n	136 (63.8)	97 (63.4)	39 (65.0)	0.827
Body mass index, kg/m^2^	23.4 ± 3.3	23.5 ± 3.2	23.1 ± 3.6	0.472
EuroSCORE (logistic)	6.3 ± 5.9	5.9 ± 6.0	7.1 ± 5.7	0.174
NYHA class ≥ 3	96 (45.1)	69 (45.1)	27 (45.0)	1.000
Diabetes, *n* (%)	33 (15.5)	22 (14.4)	11 (18.3)	0.612
Hypertension, *n* (%)	69 (32.4)	47 (30.7)	22 (36.7)	0.502
Dyslipidemia	119 (55.9)	86 (56.2)	33 (55.0)	0.873
Previous cardiac surgery	53 (24.9)	38 (24.8)	15 (25.0)	0.980
Preoperative laboratory tests				
Hematocrit, %	38.8 ± 5.8	39.6 ± 5.5	36.6 ± 5.9	0.001
Creatinine, mg/dL	1.0 ± 1.4	1.0 ± 1.6	0.9 ± 0.3	0.211
Total bilirubin, mg/dL	0.9 ± 0.6	1.0 ± 0.6	0.9 ± 0.4	0.249
Albumin, g/dL	3.8 ± 0.5	3.8 ± 0.5	3.6 ± 0.7	0.031
AST, IU/L	26.0 ± 8.4	26.2 ± 7.9	25.5 ± 9.8	0.842
ALT, IU/L	20.0 ± 11.9	20.3 ± 11.5	19.4 ± 13.0	0.103
ALP, IU/L	87.3 ± 43.4	83.8 ± 38.6	96.1 ± 53.0	0.107
Uric acid, mg/dL	6.1 ± 2.1	6.2 ± 1.9	5.9 ± 2.4	0.155
BNP, pg/mL	172.0 ± 273.9	174.2 ± 266.6	166.4 ± 294.0	0.591
C-reactive protein	0.4 ± 0.8	0.2 ± 0.3	0.7 ± 1.4	0.008
PLR	6.6 ± 6.2	4.8 ± 1.7	11.4 ± 9.9	0.001
NLR	2.2 ± 1.8	1.7 ± 0.7	3.7 ± 2.8	0.001
Preoperative echocardiographic findings				
LVEF, %	59.0 ± 7.3	58.8 ± 7.8	59.6 ± 6.2	0.765
RV dysfunction	58 (27.2)	38 (24.8)	20 (33.3)	0.182
PASP, mmHg	35.5 ± 16.4	34.6 ± 16.6	37.9 ± 15.8	0.392
Preoperative medications				
ACEI/ARB	75 (35.2)	50 (32.7)	25 (41.7)	0.866
β-blocker	66 (31.0)	47 (30.7)	19 (31.7)	1.000
CCB	60 (28.2)	38 (24.8)	22 (36.7)	0.920
Digoxin	62 (29.1)	42 (27.5)	20 (33.3)	0.289
Diuretics	156 (73.2)	109 (71.2)	47 (78.3)	0.379
Intraoperative data				
Maze operation	78 (36.6)	60 (39.2)	18 (30.0)	0.120
Valve replacement	65 (30.5)	48 (31.4)	17 (28.3)	0.661
CPB time, min	246.1 ± 53.8	247.2 ± 51.8	243.2 ± 59.0	0.102
Anesthesia time, min	354.8 ± 102.1	354.6 ± 99.1	355.3 ± 110.3	0.824

Data are expressed as number of patients (%), mean ± standard deviation, or median (first–third quartiles). Abbreviations: ACEI/ARB, angiotensin-converting enzyme inhibitor/angiotensin receptor blocker; ALT, alanine aminotransferase; ALP, alkaline phosphatase; AST, aspartate aminotransferase; BNP, brain natriuretic peptide; CCB, calcium channel blocker; CPB, cardiopulmonary bypass; EuroSCORE, European System for Cardiac Operative Risk Evaluation; LVEF, left ventricular ejection fraction; NLR, neutrophil/lymphocyte ratio; NYHA, New York Heart Association; PASP, pulmonary artery systolic pressure; PLR, platelet/lymphocyte ratio; RV, right ventricle.

**Table 2 jcm-10-04147-t002:** Postoperative outcomes for SII groups.

Variables	Total	SII <455.6 × 10^9^/L	SII ≥455.6 × 10^9^/L
N	213	153	60
Primary outcomes: composite 30-day major postoperative complications.	82 (38.5)	32 (20.9)	50 (83.3) *
30-day death	4 (1.9)	2 (1.3)	2 (3.3)
MACE	34 (16.0)	15 (9.8)	19 (31.7)
Pulmonary complications	24 (11.3)	10 (6.5)	14 (23.3)
Renal complications	16 (7.5)	9 (5.9)	7 (11.7)
Stroke	9 (4.2)	5 (3.3)	4 (6.7)
Sepsis	5 (2.3)	3 (2.0)	2 (3.3)
Multi-organ failure	5 (2.3)	4 (2.6)	1 (1.7)
Wound complications	4 (1.9)	2 (1.1)	2 (6.5)
GI complication	9 (4.2)	7 (4.6)	2 (3.3)
Secondary outcomes			
Extubation time, h	20.0 ± 50.7	15.6 ± 25.8	31.0 ± 85.7
Inotropic support, h	69.2 ± 213.8	61.3 ± 225.4	89.4 ± 181.0
Intensive care unit stay, h	114.0 ± 313.3	101.1 ± 332.5	146.9 ± 257.6
Hospital stay, days	17.5 ± 22.2	13.3 ± 18.2	28.2 ± 27.6 *
30-day readmission	14 (6.6)	10 (6.5)	4 (6.7)

Data are expressed as number of patients (%) or median (first–third quartiles). * *p* < 0.05 versus the SII < 455.6 × 10^9^/L group. Abbreviations: GI, gastrointestinal; MACE, major adverse cardiovascular events; SII, systemic immune-inflammation index.

**Table 3 jcm-10-04147-t003:** Predictors associated with composite 30-day major postoperative complications in patients undergoing isolated tricuspid valve surgery.

Variables	Univariate		Multivariable	
	Hazard Ratio (95% CI)	*p*-Value	Hazard Ratio (95% CI)	*p*-Value
SII	5.58 (3.56–8.77)	0.001	3.58 (1.62–7.95)	0.001
Sex (female)	0.97 (0.58–1.60)	0.897		
Age, years	1.03 (1.01–1.05)	0.003		
Body mass index, kg/m^2^	0.95 (0.87–1.02)	0.163		
Diabetes	1.71 (1.01–2.88)	0.046		
Hypertension	1.63 (1.05–2.54)	0.029		
NYHA	1.73 (1.12–2.67)	0.014		
EuroSCORE	1.06 (1.03–1.10)	0.001		
Preoperative hematocrit, %	0.91 (0.88–0.95)	0.001		
Total bilirubin, mg/dl	1.06 (0.70–1.60)	0.790		
Serum albumin, g/dL	0.43 (0.29–0.62)	0.001		
Creatinine, mg/dL	0.96 (0.76–1.21)	0.718		
AST, IU/L	1.00 (0.98–1.03)	0.920		
ALT, IU/L	0.98 (0.96–1.01)	0.152		
ALP, IU/L	1.01 (1.00–1.01)	0.020		
Uric acid, mg/dL	1.07 (0.95–1.20)	0.288		
BNP, pg/mL	1.00 (0.99–1.00)	0.780		
CRP	1.54 (1.31–1.82)	0.001		
PLR	1.04 (1.03–1.06)	0.001		
NLR	1.33 (1.24–1.42)	0.001		
LVEF, %	0.99 (0.96–1.02)	0.418		
RV dysfunction	1.67 (0.99–2.79)	0.053		
PASP, mmHg	0.99 (0.98–1.01)	0.494		
Use of ACEI/ARB	1.51 (0.91–2.48)	0.109		
Use of β-blocker	1.64 (0.98–2.71)	0.057		
Use of CCB	2.03 (1.29–3.16)	0.002		
Use of digoxin	1.26 (0.74–2.12)	0.391		
Use of diuretics	2.27 (1.15–4.45)	0.018		

Abbreviations: ACEI/ARB, angiotensin-converting enzyme inhibitor/angiotensin receptor blocker; ALT, alanine aminotransferase; ALP, alkaline phosphatase; AST, aspartate aminotransferase; BNP, brain natriuretic peptide; CCB, calcium channel blocker; CI, confidence interval; EuroSCORE, European System for Cardiac Operative Risk Evaluation; LVEF, left ventricular ejection fraction; NLR, neutrophil/lymphocyte ratio; NYHA, New York Heart Association; PASP, pulmonary artery systolic pressure; PLR, platelet/lymphocyte ratio; RV, right ventricle.

## Data Availability

The data presented in this study are available on request from the corresponding author (J.O.). The data are not publicly available due to ethical and institutional reasons.

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
