# Peer review of "Systemic Immune-Inflammation Index Predicted Short-Term Outcomes in Patients Undergoing Isolated Tricuspid Valve Surgery"

_jcm, 2021, doi:10.3390/jcm10184147_

Round 1
Reviewer 1 Report
The idea of considering a new inflammatory marker is intriguing. The authors correctly acknowledge the limitations of the paper; nevertheless, it is a good starting point for ideas and future larger studies.
Author Response
The idea of considering a new inflammatory marker is intriguing. The authors correctly acknowledge the limitations of the paper; nevertheless, it is a good starting point for ideas and future larger studies.
Response: We thank the reviewer for the positive comments.
Reviewer 2 Report
Systemic immune-inflammation index predicted short-term outcomes in patients undergoing isolated tricuspid valve surgery
Summary:
In this retrospective analysis, Yoon Jung Pil and colleagues investigate the association between the Systemic immune-inflammation index (SII) and acute postoperative outcomes in patients undergoing isolated tricuspid valve surgery. The authors conclude that a high SSI is associated with worse postoperative outcomes. This study has some novelty to it, but substantial revisions are required to confirm the accuracy of this conclusion. Please see my detailed comments below.
Major comments:
- The abstract needs to be revised to include additional relevant details. It is not clear how the outcome of acute postoperative complications is defined. What does it include? What time frame does it cover? How were outcomes assessed? This information is available later on in the paper but needs to be mentioned in the abstract as well.
- The link between systemic inflammation and TV surgery is not entirely clear. In the introduction, the authors explain TR and systemic inflammation separately, but fail to elaborate the link between the two. This makes it difficult for the reader to understand what knowledge gap this paper tries to fill and what question it attempts to answer. The authors should rewrite their introduction to better explain this link.
- Why were only patient undergoing TV surgery selected as opposed to all cardiac surgical patients or even all surgical patients? Systemic inflammation seems to play a major role not only in TV surgery. By including a broader cohort the authors could make more generalizable conclusions and present a much stronger study with a higher sample size.
- Definition of SII cutoff needs more explanation. The authors state they used ROC analysis to determine the optimal cutoff. Does this mean a Youden index was used? Please elaborate more on this.
- The authors should consider adding intraoperative variables to their confounder control model. Many intraoperative factors – including ventilator parameters, medications used, duration of surgery – can affect the studied outcomes. These need to be addressed to draw meaningful conclusions about the studied association.
- Results should be presented in the order of analysis: Please distinguish between primary and secondary outcomes? It is not clear in the results section which analyses and findings are primary and which are secondary. Table 2 for example shows a list a various outcomes without clear differentiation what was primary and what was secondary. In general, the results section requires more structure.
- Table 2: Please remove p values. I assume those are unadjusted p values, which are meaningless as they do not account for any confounding factors.
- P7L166-167: “This result indicates that higher baseline SII is associated with an increased risk of developing acute postoperative complications”. This statement is not justified. Again, this is solely based on unadjusted results. Please base your conclusions on adjusted analyses.
- Table 3: Please explain how these analyses were conducted. Which variables were included in the multivariable analyses? Why are multiple HRs and p values reported? I would assume the multivariable model includes all variables considered important confounders, hence resulting in one HR for the exposure. This needs to be revised and presented more clearly.
- The authors need to provide sensitivity analyses to back up their conclusions. For example, how did the results change if different cutoffs were used? How about a continuous exposure variable? Are there previously published SSI cutoffs that should be used in sensitivity analyses as well? In a retrospective analysis as this one these are crucial to solidify any conclusions. Please add these analyses.
- The discussion should more clearly dissect the association between SSI and TV surgery. The authors talk about optimal timing of surgery, yet have not studied that association. Please only draw conclusions based on your data.
Minor comments:
- The authors should be careful to correctly use the words “multivariate” vs. “multivariable”, which are not synonymous.
- Why was adult defined as >20 years instead of >18 years of age?
- P11L283: Please rephrase this to more accurately state an association, not prediction.
Author Response
To Reviewer #2
In this retrospective analysis, Yoon Jung Pil and colleagues investigate the association between the Systemic immune-inflammation index (SII) and acute postoperative outcomes in patients undergoing isolated tricuspid valve surgery. The authors conclude that a high SSI is associated with worse postoperative outcomes. This study has some novelty to it, but substantial revisions are required to confirm the accuracy of this conclusion. Please see my detailed comments below.
Response: We thank you for all your valuable and positive comments. We fully understand the potential limitations of the current study and tried our best to address each of the issues raised by the Reviewer. We hope that these revisions fulfill the Reviewer’s comments, and the specific revisions and corrections of the manuscript are as follows.
#1. The abstract needs to be revised to include additional relevant details. It is not clear how the outcome of acute postoperative complications is defined. What does it include? What time frame does it cover? How were outcomes assessed? This information is available later on in the paper but needs to be mentioned in the abstract as well.
Response 1: We thank you for this constructive comment. As recommended, we added the following content to the abstract.
“The correlation between SII and clinical outcomes was analyzed through the Cox regression analysis and the Kaplan–Meier analyses. The primary outcome was all-cause mortality and major postoperative complications within 30 days after isolated TV surgery, including major adverse cardiovascular or cerebrovascular events, renal and pulmonary complications, et al. …….with major 30-day postoperative complications……... Additionally, the Kaplan–Meier analysis revealed that the probability of undergoing major 30-day postoperative complications was significantly higher for patients with higher SII than for those with lower SII (P < 0.001). These results indicate that SII, a readily available parameter, is a significantly associated with poor outcomes in patients undergoing isolated TV surgery.”
#2. The link between systemic inflammation and TV surgery is not entirely clear. In the introduction, the authors explain TR and systemic inflammation separately, but fail to elaborate the link between the two. This makes it difficult for the reader to understand what knowledge gap this paper tries to fill and what question it attempts to answer. The authors should rewrite their introduction to better explain this link.
Response 2: We thank you for this thoughtful comment. We agree with the reviewer’ comments on that introduction does not elaborate the link between systemic inflammation and TV surgery. As the reviewer suggested, we corrected the entire introduction as follows.
“Systemic inflammation plays a central role in cardiovascular disease. Circulating neutrophils and platelets in a chronic inflammatory process and lymphocytes in subsequent immune responses are important components of atherosclerosis. Inflammation and immune-based prognostic scores, such as the neutrophil/lymphocyte ratio (NLR) and platelet lymphocyte ratio (PLR), as well as established pro-inflammatory biomarkers (leukocytosis and C-reactive protein) been shown to predict mortality and morbidity after cardiac surgery. Recently, the systemic immune-inflammation index (SII), a combination of neutrophil, platelet, and lymphocyte, has emerged as a prognostic factor in patients with cardiovascular disease. Patients with coronary artery disease, myocardial infarction, and chronic heart failure have exhibited a poor prognosis in the SII-high group. However, the association between preoperative systemic inflammation and the prognosis of isolated TV surgery has not been reported. Indeed, patients undergoing isolated TV surgery are frequently exposed to chronic inflammation because symptom-based delayed surgery, advances heart failure, and various comorbidities including diabetes, hypertension, hepatic dysfunction, and a frequent history of previous left-sided valve surgery contribute to chronic inflammation state. Also, isolated TV surgery has been known as a high-risk surgery with a 10% in-hospital mortality rate and poor long-term outcomes, regardless of the improvement of surgical techniques or postoperative management. Therefore, efforts need to be made to provide prognostic factors to predict postoperative clinical outcomes, thereby offering an appropriate patient selection for surgery.”
#3. Why were only patient undergoing TV surgery selected as opposed to all cardiac surgical patients or even all surgical patients? Systemic inflammation seems to play a major role not only in TV surgery. By including a broader cohort the authors could make more generalizable conclusions and present a much stronger study with a higher sample size.
Response 3: We appreciate this constructive comment. As you mentioned, systemic inflammation plays a major role in cardiac surgical patients, especially those with atherosclerotic cardiovascular disease. Previous studies have proved its prognostic value in patients who underwent CABG, TAVR, and valve surgeries due to infective endocarditis (Dey et al, J Cardiothorac Vasc Anesth.2021:Aug;35(8):2397-2404; Erdoğan et al., Echocardiography. 2021 May;38(5):737-744; Agus et al., J Saudi Heart Assoc. 2020 Apr 17;32(1):58-64). However, unlike left-sided valve diseases or coronary atherosclerosis, patients undergoing isolated TV surgery have various etiologies, such as purely isolated TR with aggravated chronic heart failure, concurrent left-sided valve disease, or rarely organic tricuspid valve disease. Therefore, by revealing the association between systemic inflammation and patients with various etiologies in isolated TV surgery, we could more generalize the fact that systemic inflammation is a prognostic factor not only in atherosclerotic patients but also in a more comprehensive group of heart surgery patients. We agree with your suggestion that further studies with a higher sample size are needed to confirm the prognostic role of systemic inflammation in general cardiac surgery.
#4. Definition of SII cutoff needs more explanation. The authors state they used ROC analysis to determine the optimal cutoff. Does this mean a Youden index was used? Please elaborate more on this.
Response 4: This is an excellent comment, which was appreciated by the authors. We absolutely agree with the reviewer that choosing an appropriate cut-off value for determining preoperative higher SII is important. Unfortunately, there is currently no consensus on universal cut-off values for defining higher SII in cardiac surgery patients. Previous reports of patients undergoing cardiovascular surgery analyzed receiver operating characteristic curves (Yang et al., Eur J Clin Invest. 2020;50:e13230), or used the highest quartiles (M. Xu et al., Atherosclerosis 323; 2021;20–29) to determine cutoff values and relied on calculating relative cutoff points rather than establishing an absolute cutoff. In this regard, we performed receiver operating characteristic (ROC) curves to determine the optimal cutoff value of SII for predicting major 30-day postoperative complications after isolated TV surgery with the Youden index method. As the reviewer pointed out, direct application of these values to the subjects included in our study may be considered inappropriate due to differences in underlying illness and other diverse demographic features. Therefore, when using our cut-off value for evaluating SII in patients undergoing cardiac surgery, it will be necessary to interpret cautiously. We acknowledge that this is one of the limitations of the current study and agree that this point should be clarified and discussed. Accordingly, the Materials and Methods section has been revised as follows. We appreciate again this comment from the Reviewer.
In the Materials and Methods section:
“An optimal cut-off point for SII was determined by analyzing composite of major 30-day postoperative complications. The SII cut-off value (455.6 × 109/L) was identified by analyzing the receiver operating characteristics (ROC) curve with the Youden index method between patients with and without an endpoint of the composite of major 30-day postoperative complications. This cut-off was based on previous reports.”
#5. The authors should consider adding intraoperative variables to their confounder control model. Many intraoperative factors – including ventilator parameters, medications used, duration of surgery – can affect the studied outcomes. These need to be addressed to draw meaningful conclusions about the studied association.
Response 5: We greatly appreciate your insightful comment. We completely agree that many intraoperative factors – including ventilator parameters, medications used, duration of surgery – can be associated with postoperative outcomes and should be included in the analysis. However, given the aspects of the current study, intraoperative factors (ventilator parameters, medications used) could not be identified, which is considered a major insurmountable limitation. In order to fulfill the Reviewer’s comment, we added this to the limitations as follows.
“Third, although we adjusted for various factors known to influence major 30-day postoperative complications after isolated TV surgery, the retrospective nature of this study may have masked unknown confounders or prognostic factors. Especially, inotropic agents, ventilator parameters, fluid administration, or other intraoperative events that can significantly influence major 30-day postoperative complications were not included in our analysis. Therefore, our results should be interpreted with caution.”
#6. Results should be presented in the order of analysis: Please distinguish between primary and secondary outcomes? It is not clear in the results section which analyses and findings are primary and which are secondary. Table 2 for example shows a list a various outcomes without clear differentiation what was primary and what was secondary. In general, the results section requires more structure.
Response 6: Thank you for the comment. We revised Table 2 and the result session as recommended.
#7. Table 2: Please remove p values. I assume those are unadjusted p values, which are meaningless as they do not account for any confounding factors.
Response 7: Thank you for the comment. We revised as recommended.
#8. P7L166-167: “This result indicates that higher baseline SII is associated with an increased risk of developing acute postoperative complications”. This statement is not justified. Again, this is solely based on unadjusted results. Please base your conclusions on adjusted analyses.
Response 8: Thank you for the comment. We deleted as recommended.
#9. Table 3: Please explain how these analyses were conducted. Which variables were included in the multivariable analyses? Why are multiple HRs and p values reported? I would assume the multivariable model includes all variables considered important confounders, hence resulting in one HR for the exposure. This needs to be revised and presented more clearly.
Response 9: We thank you for all your valuable and positive comments. In order to address the Reviewer’s comment, we revised the Statistical analyses and Results sessions as follows.
In the Statistical analyses;
“Univariate and multivariable analyses were performed by the Cox proportional hazards regression model. Factors related to major 30-day postoperative complications were identified through univariate analysis. The possible prognostic factors, including variables that had p-value <0.05 in univariate analysis, were incorporated into the stepwise multivariable Cox regression model.”
In the results;
Results from the univariate analysis indicated that age, EuroSCORE, preoperative hematocrit, serum albumin, C-reactive protein levels, PLR, NLR, and SII were prognostic factors for major 30-day postoperative complications (Table 3). Subsequent to a multivariable analysis of the variables, SII (hazard ratio [HR] 3.58, 95% confidence interval [CI] 1.62–7.95, P = 0.001) remained as independent prognostic factors of major 30-day postoperative complications.
#10. The authors need to provide sensitivity analyses to back up their conclusions. For example, how did the results change if different cutoffs were used? How about a continuous exposure variable? Are there previously published SSI cutoffs that should be used in sensitivity analyses as well? In a retrospective analysis as this one these are crucial to solidify any conclusions. Please add these analyses.
Response 10: We appreciate this constructive comment. In order to fulfill the reviewers' comments, based on the previously published SSI cut-offs (Yang et al., Eur J Clin Invest. 2020;50:e13230; Dey et al., J Cardiothorac Vasc Anesth . 2021 Aug;35(8):2397-2404.), we additionally performed sensitivity analyses. We revised the Statistical analyses and Results sessions as follows.
In the Statistical analyses;
“To test the robustness of our results, sensitivity analysis was performed using different cut-off values mentioned in previous studies. We used the Cox proportional hazards models and the Kaplan-Meier curves to identify relationships between prognostic factors and major 30-day postoperative complications.”
In the results;
“Sensitivity Analyses of SII Cut-Off
At each SII cut-off value selected for sensitivity analyses, we observed a significant association between preoperative higher SII and major 30-day postoperative complications. (SII 694.03×109/L as the cut-off value, HR 2.19, 95% CI 1.05–4.54, P =0.036, supplementary table 1) (SII 878.06×109/L as the cut-off value, HR 3.97, 95% CI 1.72–9.19, P =0.001, supplementary table 2). In addition, on Kaplan–Meier analysis, the probability of undergoing composite major postoperative complications within 30 days was significantly higher in patients with preoperative higher SII (log-rank test, p = 0.001, supplementary figure. 1, figure 2).”
#11. The discussion should more clearly dissect the association between SSI and TV surgery. The authors talk about optimal timing of surgery, yet have not studied that association. Please only draw conclusions based on your data.
Response 10: Thank you for the comment. You have raised an important point here. The sentences on P9L212-P9L217, P10L277- P10L278 may have misled the association between SSI and optimal timing of TV surgery. The sentences on P10L277- P10L278 were deleted to elaborate based on our findings only, not by presumption.
Therefore, we revised the sentences to clarify this point as follows.
“In this regard, several studies reported that preoperative anemia, renal/hepatic dysfunction, right ventricular dilation, and significant postoperative TR were reported to be important in determining long-term outcomes following isolated TV surgery. In addition to these prognostic factors, we found that preoperative higher SII was significantly associated with postoperative survival in TV surgery.”
Minor comments:
#12. The authors should be careful to correctly use the words “multivariate” vs. “multivariable”, which are not synonymous.
Response 12: We thank you for this comment. We agree with the reviewer’s concern. We revised the term “multivariate” to “multivariable” throughout the manuscript. We apologize for obscuring the expression.
#13. Why was adult defined as ≥ 20 years instead of >18 years of age?
Response 13: We thank you for this comment. As mentioned above, the legal adult age in Korea is 19. However, since none of the patients who underwent isolated tricuspid valve surgery during the study period were 19 years old, the patient group was determined to be 20 years or older. There appears to be no difference between study groups even if patients over 18 years of age were included in the inclusion criteria. Therefore, as mentioned by the reviewer, we modified the patient group to be 18 years of age or older, which is usually applied.
#1-14. P11L283: Please rephrase this to more accurately state an association, not prediction.
Response 14: Thank you for the comment. We revised as recommended.
“The preoperative higher SII, which may reflect the balance of the immune and inflammation pathways, was associated with an increased risk of major 30-day postoperative complications after isolated TV surgery.”

Round 2
Reviewer 2 Report
Thank you for submitting a substantially improved manuscript. I only have a few more points that should be addressed.
Abstract: Please remove "et al." in line 23 and list all included outcomes.
Some aspects of grammar and language need further corrections, especially the use of articles and pronouns. Revision by a native speaker or language editing service may be helpful.
L107: Please revise the sentence "This cut-off was based on previous reports." The method used to determine the cutoff was based on previous reports, not the actual cutoff itself as you described.
L185-187: ALP and use of CCB were also significant in univariate analyses. Were those not included in the multivariable model? If not, why not?
L210-218: Thank you for adding these sensitivity analyses. You based the 2 additional cutoffs on 2 prior publications (references 7 & 16). Other cutoffs were also used in papers you cite, e.g. references 5-8 use 4 different cutoffs ranging from 551 to 1423. Did you use those cutoffs as well? Please add results for these as well. Also, it is worthwhile looking at a cutoff based on median in your study sample. These additional sensitivity analyses will make the paper even stronger.
L64: The age cutoff is still listed as >20. Please change to >18 based on your previous response.
Author Response
To Reviewer #2
Thank you for submitting a substantially improved manuscript. I only have a few more points that should be addressed.
Response: Thank you for your valuable and positive comments.
#1. Abstract: Please remove "et al." in line 23 and list all included outcomes.
Response 1: We thank you for this constructive comment. As recommended, we added all included outcomes to the abstract.
#2. Some aspects of grammar and language need further corrections, especially the use of articles and pronouns. Revision by a native speaker or language editing service may be helpful.
Response 2: We thank you for this thoughtful comment. As the reviewer recommended, experienced scholarly writers who are native English speakers edited this manuscript, and proofread the manuscript again. In addition, we added an editing certificate.
#3. L107: Please revise the sentence "This cut-off was based on previous reports." The method used to determine the cutoff was based on previous reports, not the actual cutoff itself as you described.
Response 3: Thank you for the comment. We revised as recommended.
#4. L185-187: ALP and use of CCB were also significant in univariate analyses. Were those not included in the multivariable model? If not, why not?
Response 4: This is an excellent comment that was appreciated by the authors. We incorporated the possible prognostic factors, including variables that had p-value <0.05 in univariate analysis, into the stepwise multivariable Cox regression model. Indeed, ALP and the use of CCB were also included in the multivariable model but were not represented in the manuscript. Therefore, we added as recommended. We apologize for obscuring the expression.
“Results of the univariate analysis indicated that age, EuroSCORE, preoperative hematocrit, serum albumin, C-reactive protein, alanine aminotransferase, PLR, NLR, SII, and calcium channel blocker were prognostic factors for major 30-day postoperative complications (Table 3).”
#5. L210-218: Thank you for adding these sensitivity analyses. You based the 2 additional cutoffs on 2 prior publications (references 7 & 16). Other cutoffs were also used in papers you cite, e.g. references 5-8 use 4 different cutoffs ranging from 551 to 1423. Did you use those cutoffs as well? Please add results for these as well. Also, it is worthwhile looking at a cutoff based on median in your study sample. These additional sensitivity analyses will make the paper even stronger.
Response: We greatly appreciate your insightful comment. In order to fulfill the reviewers' comments, based on the previously published SII cut-offs (Angiology 2021, 72, 575-581; Med. Sci. Monit. 2019, 25, 9690-9701; J. Cardiothorac. Vasc. Anesth. 2021, 35, 2397-2404; Eur. Heart J. 2018, 39(suppl_1), 70; Eur. J. Clin. Invest. 2020, 50, e13230) and the median value in our study population, we additionally performed sensitivity analyses. We appreciate this comment from the reviewer.
In the Statistical analyses;
“To test the robustness of our results, sensitivity analysis was performed using different cutoff values mentioned in previous studies and the median value in our study population”
In the results;
“ (SII 1423.12×109/L as the cutoff value, HR 2.63, 95% CI 1.18–5.84, P =0.018, Supplementary Table 1) (SII 878.06×109/L as the cutoff value, HR 2.19, 95% CI 1.05–4.54, P =0.036, supplementary table 2) (SII 750.00×109/L as the cutoff value, HR 2.81, 95% CI 1.37–5.76, P =0.005, Supplementary Table 3) (SII 694.03×109/L as the cutoff value, HR 3.97, 95% CI 1.72–9.19, P =0.001, Supplementary Table 4) (SII 551.00×109/L as the cutoff value, HR 2.12, 95% CI 1.11–4.08, P =0.024, Supplementary Table 5) (SII 326.26×109/L as the median value in our study population, HR 2.13, 95% CI 1.17–3.89, P =0.014, Supplementary Table 6). In addition, on Kaplan–Meier analysis, the probability of undergoing composite major postoperative complications within 30 days was significantly higher in patients with high preoperative SII scores (log-rank test, p = 0.001, supplementary figure. 1).
#6. L64: The age cutoff is still listed as >20. Please change to >18 based on your previous response.
Response: Thank you for the comment. We revised as recommended.